# The Piezo-resistive MC Sensor is a Fast and Accurate Sensor for the Measurement of Mechanical Muscle Activity

**DOI:** 10.3390/s19092108

**Published:** 2019-05-07

**Authors:** Andrej Meglič, Mojca Uršič, Aleš Škorjanc, Srđan Đorđević, Gregor Belušič

**Affiliations:** 1Department of Biology, Biotechnical Faculty, University of Ljubljana, 1000 Ljubljana, Slovenia; mojca.ursic@danceart.si (M.U.); ales.skorjanc@bf.uni-lj.si (A.Š.); gregor.belusic@bf.uni-lj.si (G.B.); 2TMG-BMC Ltd., Štihova ulica 24, 1000 Ljubljana, Slovenia; srdjand@tmg.si

**Keywords:** muscle contraction sensor, tensiomyography, electrically stimulated skeletal muscle contraction

## Abstract

A piezo-resistive muscle contraction (MC) sensor was used to assess the contractile properties of seven human skeletal muscles (vastus medialis, rectus femoris, vastus lateralis, gastrocnemius medialis, biceps femoris, erector spinae) during electrically stimulated isometric contraction. The sensor was affixed to the skin directly above the muscle centre. The length of the adjustable sensor tip (3, 4.5 and 6 mm) determined the depth of the tip in the tissue and thus the initial pressure on the skin, fatty and muscle tissue. The depth of the tip increased the signal amplitude and slightly sped up the time course of the signal by shortening the delay time. The MC sensor readings were compared to tensiomyographic (TMG) measurements. The signals obtained by MC only partially matched the TMG measurements, largely due to the faster response time of the MC sensor.

## 1. Introduction

Skeletal muscles enable efficient movements of varying intensities and durations in different movement patterns. Objective and non-invasive evaluation of muscle contractile properties is an important tool in assessing and planning many activities related to human movement and can be applied in physiology, physiotherapy and professional sports.

Skeletal muscles’ contractile properties are extensively studied using different measuring methods [1,2,3,4]. Due to the invasiveness of direct methods for the determination of biomechanical properties in human skeletal muscles through the estimation of the muscle fibre type, muscle properties are preferably estimated through indirect measurement methods. Biomechanical properties are usually detected by measuring the mechanical power or torque about a specific joint [5]. Such measurements are non-selective, as it is generally almost impossible to measure the force or torque of an individual muscle. Other methods involve surface electromyography [6,7,8,9,10,11,12], magnetic resonance imaging [13] and ultrasound [14]. Recently, mechanomyography has been introduced. This non-invasive technique records and quantifies the low-frequency lateral oscillations produced by the active skeletal muscle fibres [15,16,17,18]. The signal can be detected by piezoelectric contact sensors, microphones, accelerometers or laser distance sensors [19]. Another suitable method to examine skeletal muscle’ properties is tensiomyography (TMG).

The tensiomyographic measuring technique (TMG) was devised to non-invasively measure the biomechanical, dynamic and contractile properties of human skeletal muscles. In TMG, muscle belly displacement is measured during the isometric muscle contraction induced externally by electrical stimulation [20,21,22]. For each twitch response, five parameters are calculated from the displacement time course. The reliability of the derived contractile parameters was extensively tested [23,24,25,26,27]. Parameters can be used to analyse the muscle tone [20,28], fatigue [29,30] and asymmetries [31]. Moreover, the measured contraction time (time between 10% and 90% of the maximum value of the muscle response) provides an important insight into the type of muscle fibre [32,33,34]. The correlation coefficient between contraction time and the percentage of type 1 muscle fibres is 0.93. The usefulness of TMG has been demonstrated in studies monitoring muscle atrophy [35], effects of different types of physical exercise [36,37,38], endurance [30] and response characteristics after intensive training periods [29].

However, TMG measurements are constrained to isometric conditions. To enable the measurements of muscle mechanical activity during free movement in non-isometric conditions a new sensor was developed [39]. 

A muscle contraction (MC) sensor was first introduced in 2011 [39]. This is a surface sensor, which transduces the deformation of the contracting muscle into resistance with four piezo-resistors connected in a Wheatstone bridge. During the measurement, the sensor is fixed on the skin surface above the muscle belly while the sensor tip applies pressure and causes an indentation of the skin and intermediate layer directly above the muscle and muscle itself. The force on the sensor tip is then measured [39]. In the study by Đorđević et al. [39] a high correlation between elbow peak flexion force at five target levels of maximum voluntary contraction and MC sensor measured peak tension was observed for the biceps brachii muscle. Dynamic properties of the sensor were further investigated in the time domain in the same muscle [40]. Statistical correlation between MC signal and force was very high at 15° (r = 0.976) and 90° (r = 0.966) elbow angle. The MC sensor dynamically follows the contractions and is thus suitable for the dynamic measurement during voluntary free motion in cases where direct force cannot be measured. Due to differences in skin and subcutaneous tissue mechanical properties and thickness, the MC sensor cannot be used for measurements of the absolute value of muscle force without proper calibration [40]. A strong correlation between MC sensor signal and muscle force as well as a close-to-zero delay between the peak muscle force and MC sensor signal was shown in upper trapezius muscle during a low-severity frontal impact [41]. Mohamad et al. [42] evaluated the sensor using functional electrical stimulation. A strong linear correlation between MC sensor measurements and dynamometer isometric knee torque suggested that the MC sensor is able to detect different contraction levels and fatigue of the rectus femoris muscle among individuals with spinal cord injury [42].

The aim of this study was to evaluate the MC sensor function during stimulated twitch contractions, to measure the effect of the sensor tip depth on the obtained signal and to compare the results with the parallel tensiomyographic measurements. 

## 2. Materials and Methods

### 2.1. Subjects

Ten female volunteers, ranging between 18 and 24 years of age participated in the study. They were all athletes in various dance categories (hip hop, modern, ballet), 159 cm–174 cm tall, weighing 48–63 kg. Subcutaneous fat, measured using a skin fold caliper, ranged: 3.4–5.3 mm at the biceps, 20.0–20.2 mm at the thigh, 3.2–14.4 mm at the calf and 6–8.3 mm at the lower back site. All subjects were healthy and had no known neuromuscular or musculoskeletal disorders at the time of the study. All subjects gave their informed consent for inclusion before they participated in the study. The study was conducted in accordance with the Declaration of Helsinki, and the Republic of Slovenia National Medical Ethics Committee approved the experimental procedures.

Seven different muscle pairs (left and right muscle of each volunteer) were analysed: m. vastus medialis (VM), m. vastus lateralis (VL), m. rectus femoris (RF), m. tibialis anterior (TA), m. gastrocnemius medialis (GM), m. biceps femoris (BF) and m. erector spinae (ES). During the measurements the subject was in a prone or supine position, depending on measured muscle. Measurements were performed under static, relaxed conditions. The measured limb was positioned on a triangular wedge foam cushion to keep the knee joint in natural physiognomic position—flexed for 20°. Special care was taken during the measurements of m. erector spinae to prevent breathing-movement artefacts. 

### 2.2. Tensiomyographic (TMG) Measurements

We first performed the TMG measurements. A digital displacement sensor (TMG-BMC d.o.o., Ljubljana, Slovenia) was placed perpendicular to the tangential plane on the largest area above the muscle belly. The surface area of the dome shaped sensor tip is 35 mm^2^. The force applied to the relaxed muscle was 1.5 N and the indentation was approximately 6 mm. The measuring point was anatomically determined on the basis of the anatomical guide for electromyographers [43] and marked with a dermatological pen. Self-adhesive bipolar electrodes (Axelgaard Manufacturing Co., Fallbrook, CA, USA) were placed symmetrically 2–4 cm distal and proximal to the sensor tip. For stimulation we used 1 ms long square pulse, the amplitude was progressively increased (40–100 mA) to obtain a maximal response. 

### 2.3. MC Measurements

The MC sensor principle was introduced by Đorđević et al. [39]. In brief, the sensor consists of a supporting part (650 mm^2^ surface area) made of an elliptically shaped carbon fibre reinforced epoxy polymer. An incision forms a tonguelet to which the sensor tip is attached. The surface area of the sensor tip applying pressure to the skin is 56 mm^2^. A piezo-resistive strain gauge is attached at the root of the tonguelet. The strain is proportional to the force acting on the sensor tip. Muscle contraction produces tension which causes subcutaneous tissue and skin to press on the sensor tip. The MC sensor was produced by TMG-BMC d.o.o. (Ljubljana, Slovenia).

The sensor was attached to the skin through the supporting part using double-sided adhesive patches (Figure 1). The sensor tip was placed on the mark made on the skin during the TMG measurements. The depth of the sensor tip indented in the skin fold was set to 3, 4.5 and 6 mm. The depths of the sensor tip were chosen according to the TMG measurements and MC sensor technical properties: 3.5 mm is the minimum depth limited by the MC sensor tip size; 6 mm is comparable to the indentation of the skin and subcutaneous tissue in TMG measurements; 4.5 mm was chosen between these two values for comparison. At greater depths, the force on the sensor tip during muscle contraction was such that the sensor detached from the skin. The sensor was designed to allow changing of the tip depth without removing the sensor. The stimulation electrodes remained attached to the skin when we switched from the TMG to the MC sensor. Time between TMG and MC measurements was around two minutes. The muscle contraction was triggered with the same current as in TMG measurements. Two successive measurements were made for each muscle at each depth.

The MC signal was sampled at 10 kHz using a 24-bit resolution, 25 mV/V NI 9237 module (National Instruments, Austin, TX, USA). The sensor output response (in mV/V) was recalculated into force using the calibration curve obtained by suspending weights (1, 2, 5, 10, 20, and 50 g) on the tonguelet. The process of calibration, timeline of MC output response at various weights and sensitivity graph are shown in Đorđević et al., 2011 [39]. The dependence between force and sensor output was linear.

MATLAB (MathWorks, Natick, MA, USA) was used for data processing. Figures and Tukey’s test, used for the time parameter multiple comparisons were done in Prism 8.0 (GraphPad Software, San Diego, CA, USA).

## 3. Results

The typical MC sensor recording is presented in Figure 2. The time course of muscle contraction shows a rapid onset due to the action of the fast contractile elements, resulting in the first peak within a few tens ms after the stimulus. The slower contractile elements reach the peak activity within >100 ms after the stimulus. The subsequent relaxation follows a simpler time course. Five parameters were extracted from the measured response of the muscle belly to the electrical stimulus: 

Dm—maximal force.

Td—delay time (from 0 to 10% of Dm).

Tc—contraction time (from 10% of Dm to 90% of Dm or 90% of the first explicit peak amplitude. The criterion for identifying the first peak in the couple was a drop in the signal for 2%.

Ts—sustain time (signal >50% of Dm).

Tr—relaxation time (signal drops from 90 to 50% of Dm).

Measurements of seven different muscles made with TMG and MC sensor at three different sensor tip depths were compared. Signals from a single muscle are presented in Figure 3a–d; signals from all muscles are presented in Appendix A. To compare the signals from the two sensors, Pearson correlation coefficient was calculated between the synchronized data points from the two measurements in the same muscle (Table 1). The average correlation coefficient *R* between TMG and MC signal was at the different depths of sensor tip (3, 4.5, 6 mm): *R*_(3 mm)_ = 0.72, *R*_(4.5 mm)_ = 0.76, *R*_(6 mm)_ = 0.76. The only exception was the muscle erector spinae, where the correlation coefficient was considerably lower and surprisingly the highest at the shallowest depth *R*_(3 mm)_ = 0.62.

The increase in the depth of the sensor tip resulted in an increase of the amplitude (Dm) of the measured signal (Figure 3b–d). The increase was ×1.7–×3.0 and ×2.0–×4.8 when the depth of the tip was increased from 3 to 4.5 and 6 mm, respectively. The lowest increase was in the muscle tibialis anterior (×1.7 and ×2.0). 

The time courses of the muscle response to electrical stimulus vary depending on the muscle and the chosen method (Figure 3e). In general, we distinguish two basic types: single and double peak signal. The two peaks were usually better resolved in recordings with the MC sensor. However, the frequency of occurrence of double peaks was for some muscles almost identical in both measuring techniques. In measurements from gastrocnemius medialis and vastus medialis, the double peaks were present in 95% of TMG recordings and 95–100% (depending on the sensor depth) recordings with MC sensor. However, in vastus lateralis, biceps femoris and rectus femoris, the double peaks were present in only 30% of TMG and in 85–95% of MC recordings. The tibialis anterior muscle was an exception with the double peaks occurring in only 25% of recordings, regardless of the sensor type. 

The depth of the MC sensor tip had little impact on the Tc, Ts and Tr (with the exception of Tr parameter in vastus medialis) time course parameters. Statistically significant differences between different tip depths were observed in the parameter Td in four out of seven muscles (exceptions were gastrocnemius medialis, erector spinae and biceps femoris). The Td time in vastus medialis, rectus femoris, vastus lateralis and tibialis anterior shortened with the increasing tip depth on average for 19.3% and additional 3.0% for an increase from 3.0 to 4.5 and 6 mm. 

When compared to TMG measurements, Td times were shorter when measured with MC sensor (Figure 3e,f; Figure 4a,b) with the exception of erector spinae where no statistically significant differences were found between TMG and MC measurements. On average Td was shorter for 20.3 to 26.5% at the middle depth of the sensor tip compared to TMG. 

Similar results were obtained for the parameter Tc with 20.4 to 33.0% shorter times measured with MC sensor on middle depth than with TMG (Figure 3e,f). Longer times were measured on vastus lateralis and erector spinae muscle (for 26.6 and 34.5% respectively) although the difference was not significant. 

Statistically significant differences in Ts times were measured in three muscles—vastus lateralis, erector spinae (between TMG and all three depths) and biceps femoris (between TMG and MC at 6 mm depth). While the times measured with MC sensor were shorter in biceps femoris and erector spinae (16.0% and 27.3%) they were longer in vastus lateralis (70.5%, 60.8% and 60.7% for 3, 4.5 and 6 mm MC sensor tip depth).

The least differences between the two methods were found for the Tr parameter (Figure 4d). The significant difference was found only in vastus medialis, erector spinae (between TMG and MC at 3 mm depth) and biceps femoris (between TMG and MC at 6 mm depth). 

## 4. Discussion

Two sensors, MC and TMG, were used to evaluate the contraction in different types of muscles—short, long, one and two jointed, phasic and postural. The lowest correlation coefficient between TMG and MC sensor signal was measured in erector spinae. In contrast to other measured muscles that are simpler, resembling a spindle interconnecting two parts of a limb, erector spinae is a bundle of muscles and tendons varying in size and structure at different parts of the vertebral column. In the lumbar region, where the measurements were made, it is a large, thick fleshy mass. Its contraction results in a muscle displacement with a less pronounced muscle belly. As the MC sensor moves together with the muscle, the pressure on the sensor tip is in such case smaller resulting in lower signal. Stimulation of erector spinae increases the back lordosis, affecting TMG measurements, as the sensor does not move together with the back. Nevertheless, the time parameters between the two methods for this muscle are fairly well matched with the exception of Tc times, which are shorter in MC measurements.

In other muscles, the correlation between TMG and MC measurements was between 0.66 and 0.84, depending on the MC sensor tip depth. We assume that the signals differed in the shape of their time course mainly due to the different mass of the two sensors. The MC sensor, which is much lighter and has a lower inertia can, therefore, respond more rapidly to changes in the muscle tension, and detects smaller differences. The faster responsively of the MC sensor is also visible in shorter Td and Tc times and in higher number of double peaks which are also more pronounced (Figure 3e,f). The variability in presence and distinctiveness of double peaks caused substantial scattering of the Tc values, especially in MC measurements. To avoid this we measured maximal slopes. A linear function was fitted to 10 ms segments of the signal, normalised to maximal displacement or force. The maximal slopes of the signal (Figure 5) calculated show similar trends as in Tc times (Figure 4b) with much smaller standard deviations.

The depth of the MC sensor tip influenced mostly the amplitude of the signal. In all measured muscles, the force increased with the increasing tip depth. A relatively small increase in the tibialis anterior was due to a thin layer of soft tissue between the sensor and bone. Strong curvature of this part of the leg also prevented the firm attachment of the MC sensor to the skin especially at the maximal sensor tip depth. Upon removing the MC sensor, we discovered that the bond between skin and sensor in tibialis anterior has been loose on several occasions with the 6 mm MC sensor depth. In such cases, we reattached the sensor and repeated the measurement. In general, out of the three preset values the medium depth of 4.5 mm is the best compromise between the signal amplitude and firm contact.

In general, the signals from the MC sensor had shorter Td and Tc parameters (Figure 4) and larger slopes (Figure 5) than the signals from the TMG sensor, which we attributed to the faster response of the MC sensor. To verify this assumption, both sensors were tested with a mechanical actuator and laser vibrometer. Indeed, the MC sensor was capable of following a 10 ms ramp displacement while the TMG sensor had substantially slower response (Appendix A). Even though the MC sensor is slightly less robust than the TMG sensor it appears to be more suitable for the measurements in fast muscles. For more comprehensive comparison of mechanical properties of both sensors further measurements need to be done.

## 5. Conclusions

We measured a stimulated isometric contraction of seven muscle pairs (6 on leg and 1 on back) in ten healthy volunteers aged 18 to 25. Firstly, a tensiomyographic (TMG) measurement was done, with the sensor perpendicular to the skin overlying the muscle belly and self-adhesive stimulating electrodes 2–4 cm distally and proximally of the sensor tip. Then, at the site of the placement of the TMG sensor, a MC sensor was attached using double-sided adhesive patches. The stimulation parameters were the same as for the TMG measurements. Sensor tip depth was adjusted to 3, 4.5 and 6 mm. For each measurement the delay (Td), contraction (Tc), sustain (Ts) and relaxation (Tr) time was extracted from the measured response of muscle belly on electrical stimulus. The average correlation coefficient between TMG and MC measurement for vastus medialis, rectus femoris, vastus lateralis, gastrocnemius medialis, and biceps femoris at all three depths of sensor tip was 0.74. The exception was the erector spinae muscle with lower average correlation coefficient (r = 0.52). The depth of the sensor tip affected the measured force but had little influence on the normalized time course, except on the parameter Td, which decreased with the increasing tip depth. The measured parameters obtained with the MC sensor were partially in agreement with the tensiomyographic measurements. Td and Tc times were on average shorter for 24.2% and 25.7% respectively. The exceptions are Td times in erector spinae and Tc times in erector spinae and vastus lateralis where the times were shorter in TMG. Ts and Tr times were more comparable between MC sensor and TMG. Statistically significant differences in Ts times between MC and TMG measurements were obtained in biceps femoris, erector spinae and vastus lateralis while Tr times were different in biceps femoris, erector spinae and vastus medialis although only between TMG and MC at 3.5 and 6 mm tip depth.

We can conclude that the MC sensor enables the measurement of the mechanical muscle properties. However, the extracted time parameters cannot be directly compared to the ones measured with the TMG. The MC sensor can respond more rapidly and consequently allowing a better separation between fast and slow fibres in the muscle. Since muscle architecture changes during contraction are complex [44] further studies are needed to understand all the components of the signal and the origin of differences between signals from both sensors.

## Figures and Tables

**Figure 1 sensors-19-02108-f001:**
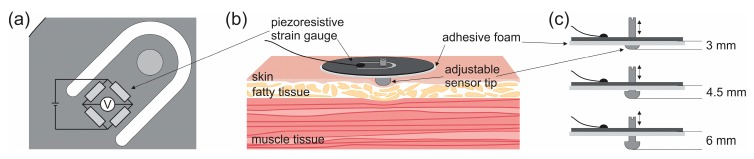
Muscle contraction (MC) sensor design and placement. (**a**) Four piezoresistors, connected in Wheatstone bridge, were attached at the root of the tonguelet. Piezoresistors contact pads and cable interface were connected with microelectronic golden wires and covered with epoxy casting compound for protection. (**b**) The sensor was attached to the skin through the supporting part using double-sided adhesive patches. (**c**) The depth of the sensor tip was set to 3, 4.5 and 6 mm. The sensor compressed the skin and subcutaneous tissue, exerting pressure on the measured skeletal muscle.

**Figure 2 sensors-19-02108-f002:**
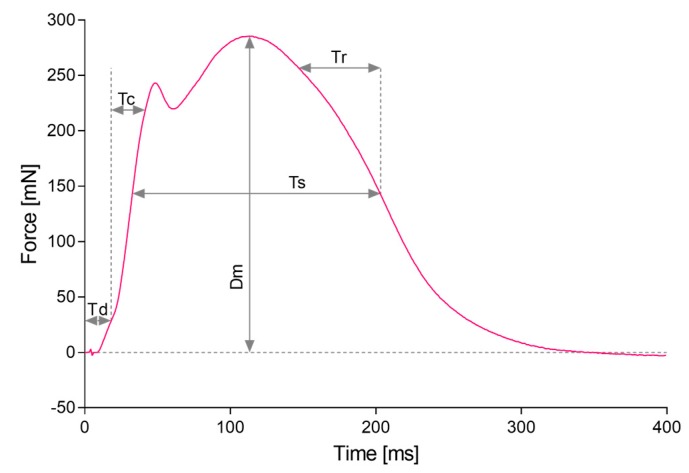
Typical MC sensor recording. Parameters extracted from measured response: Td—delay time, Tc—contraction time, Ts—sustain time, Tr—relaxation time, Dm—maximal force. Zero time corresponds to the start of electrical stimulus.

**Figure 3 sensors-19-02108-f003:**
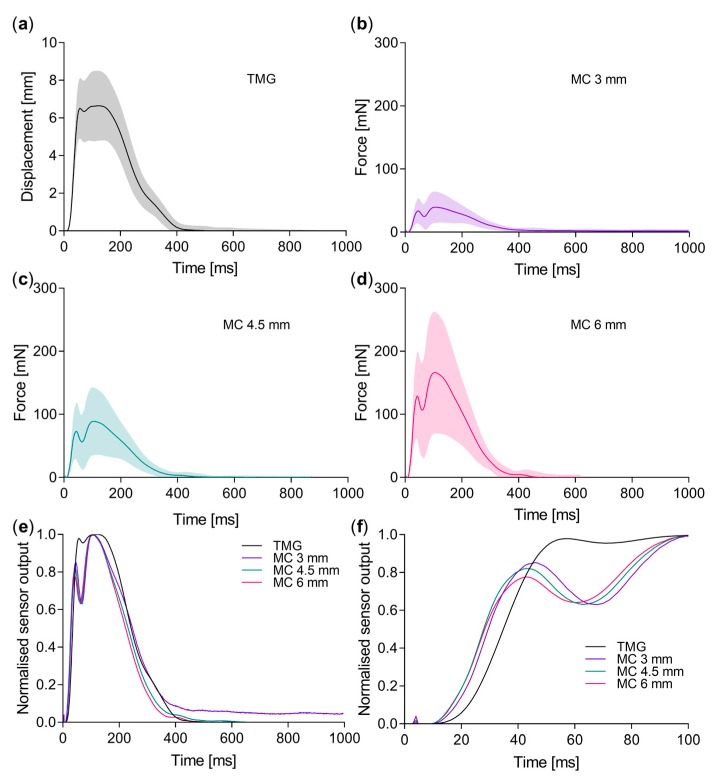
Response of vastus medialis to twitch stimulation measured with tensiomyographic (TMG) (**a**) and MC sensor at three tip depths: 3 mm (**b**), 4.5 mm (**c**) and 6 mm (**d**). Solid line, average values (*n* = 20, left and right muscles of 10 subjects were measured); envelope, SD. Average signal from both sensors has been normalised to maximum (**e**). (**f**) shows first 100 ms of the normalised signals from panel e. In some recordings, a small peak appears 4 to 5 ms after the start of the stimulus. The short duration of the peak (less than 0.5 ms) indicates non-physiological origin.

**Figure 4 sensors-19-02108-f004:**
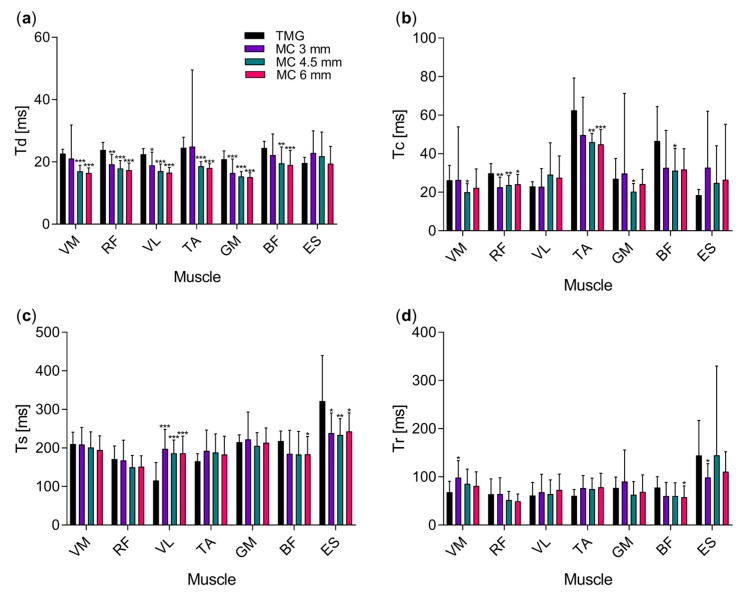
Comparison of four different temporal parameters extracted from responses of muscle belly to electrical stimulus, measured with TMG and MC. (**a**) Delay time, Td; (**b**) contraction time, Tc; (**c**) sustain time, Ts (**d**) relaxation time, Tr. Measurements with TMG are compared to those with MC sensor at three different depths (3, 4.5 and 6 mm). Asterisks indicate the level of statistical significance (* *P* < 0.05, ** *P* < 0.01, *** *P* < 0.001). Mean values and standard deviations are shown.

**Figure 5 sensors-19-02108-f005:**
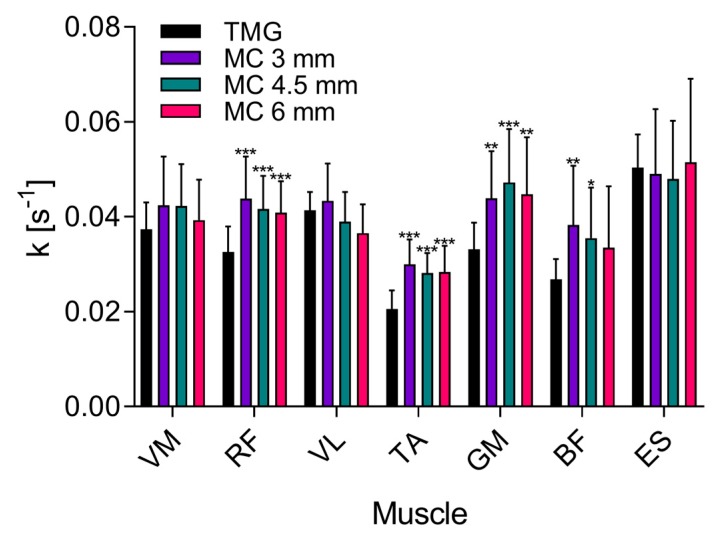
Maximal slopes of signals. Measurements with TMG are compared to those with MC sensor at three different depths (3, 4.5 and 6 mm) and their maximal slopes are averaged. Mean values and standard deviations are shown.

**Table 1 sensors-19-02108-t001:** Correlation coefficients *R* between tensiomyographic (TMG) and muscle contraction (MC) sensor measurements at the three different sensor tip depths (3, 4.5 and 6 mm). The highest coefficients in each muscle are bolded.

Muscle	*R* _(3 mm)_	*R* _(4.5 mm)_	*R* _(6 mm)_
vastus medialis	0.78	0.83	**0.84**
rectus femoris	0.72	0.80	**0.84**
vastus lateralis	0.68	0.74	**0.76**
tibialis anterior	0.74	**0.77**	0.71
gastrocnemius medialis	0.68	**0.76**	0.72
biceps femoris	**0.69**	0.68	0.66
erector spinae	**0.62**	0.49	0.44

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
