# Peer review of "The Piezo-resistive MC Sensor is a Fast and Accurate Sensor for the Measurement of Mechanical Muscle Activity"

_sensors, 2019, doi:10.3390/s19092108_

Round 1

Reviewer 1 Report

Brief summary

This article presents a comparative study of two different sensors (MC and TMG) able to measure mechanical features of a contracting muscle. The TMG sensor measures skin displacements while the MC sensor measures local force exerted by the muscle contraction. The study presented in this manuscript is interesting and innovative. However, some aspects should be addressed, in order to make the article suitable for publication.

Broad comments

Comparison between the two sensors was based on electrically stimulated contraction of specific muscles, but the data were not-simultaneously acquired.

The Authors do not provide any quantitative data about the dynamic responses (or the bandwidth) actually offered by the two sensors (MC and TMG).

The comparison between muscle-related displacement and force raises some doubts. The mechanical properties of the tissues are not reported; the viscoelastic properties of skin, fat and muscle complicates the scenario. What relationship between displacement and force was assumed?

The Authors cite the term mechanomyography. In the general meaning the term mechanomyography (MMG) refers to a technique by which the mechanical activity of muscle is detected using specific transducers to record the little muscle surface oscillations due to mechanical activity of the motor units. MMG can be detected using piezoelectric contact sensors, microphones, accelerometers, laser distance sensors and, recently, force sensitive resistor. (see:  Islam MA, Sundaraj K, Ahmad RB, Ahamed NU. Mechanomyogram for muscle function assessment: a review. PLoS One. 2013;8(3):e58902. doi:10.1371/journal.pone.0058902). Alternatively, TMG and MC sensors measure gross muscle expansions. Furthermore, this manuscript presents TMG and MC sensors to record electrically stimulated muscle contraction (no variable motor unit recruitment as in physiological contraction). Therefore, the use of the term mechanomyography (MMG) does not seem appropriate for the sensors presented or, at least, can be confusing for the readers.

No alternative sensors were cited nor compared.

These points should be clearly addressed and specified in the manuscript.

Specific comments

ABSTRACT

Page 1, line 13: “A piezo-resistive mechanomyographic muscle contraction (MC) sensor”. Please refer to the general comment. If not strictly necessary, please avoid the term “mechanomyographic” or justify its use.

Page 1, line 21: “largely due to the faster response time of the MC sensor”. Please, refer to the general comments. This statement should be demonstrated or should be removed from the abstract.

INTRODUCTION

Page 1, line 29: “Skeletal muscles’ contractile properties are extensively studied using different measuring methods.” As authors did not report any reference for such methods, they are recommended to cite the following articles:

- D. Esposito, E. Andreozzi, A. Fratini, G.D. Gargiulo, S. Savino, V. Niola, P. Bifulco. A Piezoresistive Sensor to Measure Muscle Contraction and Mechanomyography. Sensors 2018, 18(8), 2553; doi:10.3390/s18082553

- Zhou, B.; Sundholm, M.; Cheng, J.; Cruz, H.; Lukowicz, P. Measuring muscle activities during gym exercises with textile pressure mapping sensors. Pervasive Mob. Comput. 2017, 38 Pt 2, 331–345.

- Jung, P.G.; Lim, G.; Kim, S.; Kong, K. A Wearable Gesture Recognition Device for Detecting Muscular Activities Based on Air-Pressure Sensors. IEEE Trans. Ind. Inform. 2015, 11, 485–494.

- Bansal, A.K.; Hou, S.; Kulyk, O.; Bowman, E.M.; Samuel, I.D.W. Wearable Organic Optoelectronic Sensors for Medicine. Adv. Mater. 2015, 27, 1521–4095.

- Han, H.; Kim, J. Active muscle stiffness sensor based on piezoelectric resonance for muscle contraction estimation. Sens. Actuators A 2013, 194, 212–219.

- Guo, J.Y.; Zheng, Y.P.; Xie, H.B.; Chen, X. Continuous monitoring of electromyography (EMG), mechanomyography (MMG), sonomyography (SMG) and torque output during ramp and step isometric contractions. Med. Eng. Phys. 2010, 32, 1032–1042.

- Kenney, L.P.; Lisitsa, I.; Bowker, P.; Heath, G.H.; Howard, D. Dimensional change in muscle as a control signal for powered upper limb prostheses: A pilot study. Med. Eng. Phys. 1999, 21, 589–597.

Page 1, lines 36-40 and particularly lines 39-40: “A technique resembling the more established mechanomyography is tensiomyography (TMG).” Please refer to general comments on mechanomyography. The authors are suggested to rearrange this sentence for the sake of clarity. The paper “Mechanomyogram for muscle function assessment: a review” clearly separates the mechanomyography (MMG) from tensiomyography (TMG).

Page 2, line 54: “a new member of mechanomyography-based sensors was developed”. Again, refer to general comments on mechanomyography.

MATERIALS AND METHODS

Page 2, line 87: “Seven different muscle pairs”. Can you specify what is intended for pairs?

Page 3, line 95:  Please, specify the area and the shape of the “digital displacement sensor” applied to the muscle. Provide also some information about how the contact between TMG sensor and skin was obtained and secured and the amount of deformation of the skin (was the deformation comparable with that of the MC sensor?)

Page 3, line 100: Please, specify the electrical stimulation pulse shape (e.g. square wave)

Page 3, line 114: “when we switched from the TMG to the MC sensor.” How long was the time delay between the two type of measurement?

Page 3, line 119 “The sensor output response (in mV/V) was recalculated into force using the calibration curve obtained by suspending weights (1, 2, 5, 10, 20, and 50 g) on the tonguelet” Can you report a graph of the calibration? Was the relationship linear? How accurately the experimental data fitted the relationship you used? Can you specify how weights were applied on the tonguelet?

RESULTS

Page 3, lines 128-129: “Measurements of seven different muscles made with TMG and MC sensor at three different sensor tip depths were compared (Figure 3a, b, c, d).” Figure 3 report data from only one muscle. Can you add other data for sake of completeness? At least, the recordings acquired from the erector spinae, which scored the worst correlation should be added.

Page 4, line 135: “Dm – maximal displacement amplitude”: it is improper to use the term displacement and measure it in Newton (see y-axis of Figure 2). Please correct it.

Page 4, Figure 2: Please, specify if time=0 corresponds to the electrical stimulus.

Page 4, line 152: “The depth of the sensor tip increased the amplitude (Dm) of the measured signal” can you report also the amount of force recorded by the sensor when the muscle was at rest (the sensor mechanical pre-stretching corresponding to the depth of the sensor tip)?

Page 4, line 156: Figure 3f . Can you provide some explanation about the small peaks occurring at about 5 milliseconds? Are they artifacts?

Page 5, line 157: “Black curve, average values (n=20)” Please, specify if the average involved all 10 subjects (2 recording per subject).

Page 5, lines 161-162: “In general, we distinguish two basic types: single and double peak signal. The two peaks were usually better resolved in recordings with the MC sensor.” Can you report a recording showing a single peak signal as an example?

Page 5, line 169: “The depth of the MC sensor tip had no impact on the Tc, Ts and Tr”. “no impact” seems too strong!  You may use ‘little impact’ or other smoother expression.

DISCUSSION

Page 6, line 192: Figure 4. please specify what the bar and the segment represent in the plots (mean and SD?)

Page 7, line 206-207: “Nevertheless, time parameters between the two methods for this muscle are fairly well matched.” This assumption it is not fully supported by data presented in Figure 4. Please modify the statement.

Page 7, lines 209-213: “We assume that the signals differed in the shape of their time course mainly due to the different mass of the two sensors. The MC sensor, which is much lighter and has a lower inertia can, therefore, respond more rapidly to changes in the muscle tension, and detects smaller differences. The faster responsivity of the MC sensor is also visible in shorter Td and Tc times and in higher number of double peaks which are also more pronounced (Figure 3e, f).” Please, refer to the general consideration. Can you specify more detailed data (e.g. the mass of the two sensors, the mechanical parameters of the sensor-skin interface, the mechanical properties of patients’ tissues, etc.)? Otherwise, clearly state their unavailability. Information about the bandwidth of the two sensors is also of interest.

Page 7, lines 218-219: “A linear function was fitted to 10 ms segments of the signal, normalized to maximal displacement or force.” The definition of the “Maximal slopes of signals” should be reported in the manuscript text and not only in the caption of Figure 5.

Page 7, lines 225-226: “Upon removing the MC sensor, we discovered that the bond between skin and sensor has been loose on several occasions.” Can you be more precise? How many occasions? Do you mean that the bigger is the depth of sensor tip the more likely the sensor is prone to detachment? Did it happen for all the muscles? Were the signal recordings related to these detachment events disregarded?

Page 8, lines 236-237: “The average correlation coefficient between TMG and MC measurement was 0.74 with the exception of erector spinae muscle (r=0.52).” was the average correlation coefficient computed including data of the erector spinae? Please specify.

Author Response

We are grateful for the careful examination of our manuscript and for constructive comments. Our replies follow in the Word attachment.

Reviewer 2 Report

In this paper, the piezo-resistive MC sensor was used to measure the mechanical muscle activity. The manuscript is interest to readers. However, there are some problems in the manuscript.

1.     Some expressions are confusing. What are D1, D2, and D3 in the Figure 3(e), Figure 3(f) and Figure 4. The length of the sensor tip indented in the skin fold and the tip depth are confusing. Why not mark all parameters in Figure 1.

2.     A figure about the measurement circuit for the piezo-resistive MC sensor is good for understanding.

3.     Please check the manuscript for linguistic errors(e.g. for 20.3 to 26.5 %). Other formats (e.g. variable italic (Td, Tc, Ts, Tr, Dm)).

4.     The conclusion about medium depth of 4.5 mm could be proven to be the best for only three preset values.

I would suggest to revise the paper to make it proper representative of the presented work.

Author Response

(The authors gave the same response as above.)

Round 2

Reviewer 1 Report

The authors of the manuscript “The piezo-resistive MC sensor is a fast and accurate sensor for the measurement of mechanical muscle activity” (MS#474399) have substantially addressed the Reviewers’ comments and appropriately amended the manuscript, which results improved. However, Authors have not fully justified their statements about the bandwidth of the sensors yet (the complexity of such mechanical tests are comprehensible). This information does not substantially affect the presented results. Therefore, I suggest being more cautious about the assumptions and statements about this issue (still present in the amended manuscript), postponing a more complete analysis to future studies.

I suggest to accept the manuscript for publication after these few modification.

Author Response

We are grateful for constructive additional comments. Our replies follow in the attachement.

Reviewer 2 Report

In this paper, the piezo-resistive MC sensor was used to measure the mechanical muscle activity. The manuscript is interest to readers. However, there are some problems in the manuscript.

1.     Please check the manuscript for linguistic errors(e.g. for 20.3 to 26.5 %). Other formats (e.g. variable italic (Td, Tc, Ts, Tr, Dm)).

2.     The conclusion about medium depth of 4.5 mm should be proven to be the best for only three preset values.

3.     Why the depth of the sensor tip indented in the skin fold was set to 3, 4.5 and 6 mm? Why not 4.4 or 4.6?

I would suggest to revise the paper to make it proper representative of the presented work.

Author Response

(The authors gave the same response as above.)
